# Urban and Rural Disparities in Pneumococcal Carriage and Resistance in Jordanian Children, 2015–2019

**DOI:** 10.3390/vaccines9070789

**Published:** 2021-07-14

**Authors:** Adnan Al-Lahham, Nashat Khanfar, Noor Albataina, Rana Al Shwayat, Rawsan Altwal, Talal Abulfeilat, Ghaith Alawneh, Mohammad Khurd, Abdelsalam Alqadi Altamimi

**Affiliations:** 1Department of Biomedical Engineering, School of Applied Medical Sciences, German Jordanian University, Amman 11180, Jordan; Noor.albataina@gju.edu.jo (N.A.); R.Alshwayat@gju.edu.jo (R.A.S.); R.Altwal1@gju.edu.jo (R.A.); T.abulfeilat@gju.edu.jo (T.A.); G.Alawneh1@gju.edu.jo (G.A.); M.khurd@gju.edu.jo (M.K.); A.Alqadialtamimi@gju.edu.jo (A.A.A.); 2Pediatric Clinic, Khalda, Salim Khouri Str. 48, Amman 11953, Jordan; nashatkhanfar@yahoo.com

**Keywords:** *Streptococcus pneumoniae*, carriage, resistance, PCVs, urban, rural

## Abstract

Background: A pneumococcal carriage surveillance study took place examining Jordanian children in urban and rural areas in the period 2015–2019. Objectives: To determine urban and rural differences in pneumococcal carriage rate, resistance, and serotypes among healthy Jordanian children from Amman (urban) and eastern Madaba (rural). Methods: Nasopharyngeal swabs (NP) were taken from 682 children aged 1 to 163 months. Pneumococcal identification, serotyping, and resistance were performed according to standard method. Results: The number of cases tested for Amman was 267 and there were 415 cases tested for eastern Madaba. Carriage rate for eastern Madaba was 39.5% and 31.1% for Amman. Predominant serotypes for eastern Madaba and Amman were 19F (21.3%; 15.7%), 23F (12.2%; 9.6%), 14 (6.7%; 2.4%), 19A (4.9%; 2.4%), and 6A (5.5%; 3.6%). Resistance rates for eastern Madaba and Amman were as follows: penicillin (95.8%; 81.9%), clarithromycin (68.9%; 59.0%), clindamycin (40.8%; 31.3%), and trimethoprim-sulfamethoxazole (73.2%; 61.4%). Coverage of PCV7, PCV13, and the future PCV20 for Amman was 42.2%, 48.2%, and 60.2%; for eastern Madaba, coverage was 50.0%, 62.2%, and 73.2%, respectively. In Amman 25.8% of children received 1–3 PCV7 injections compared to 1.9% of children in eastern Madaba. Conclusions: There were significant differences in carriage, resistance, and coverage between both regions. The potential inclusion of a PCV vaccination program for rural areas is essential.

## 1. Introduction

*Streptococcus pneumoniae* is an infectious agent that causes invasive and non-invasive infections, especially in low-income countries with low or minimum vaccination rates [1,2,3]; it is an opportunistic human-adapted pathogen transmitted by the nasopharynx, which is an ecological reservoir for this bacterium, mainly during the first few months of life [4]. This NP carriage present in healthy children in rates between 20 and 40%, and it is influenced by different risk factors, including day care center attendance, the size of day care center, living in close contact with siblings, living in urban or rural areas, previous antibiotic consumption within one month before sample collection, cigarette smoking within family members, overcrowding in households, and socioeconomic factors [5,6,7]. Furthermore, this NP carriage is subject to genomic changes during natural colonization [8], and is acquired at approximately 4–6 months of age [1,9]. Unlike in children, the carriage is rarely detected in older populations [10]. Pneumococcal carriage prevalence is used to estimate the potential for the use of pneumococcal conjugate vaccines (PCVs) in reducing the vaccine-type isolates of *S. pneumoniae* [3], and can therefore indirectly determine the impact of PCVs on disease [11]. Annual death estimates from the World Health Organization in relation to pneumococcal infections is more than 1.2 million children, as estimated in 2006 [12,13]. Therefore, introduction of PCVs is necessary, as they have proven to be efficient in decreasing NP carriage and resistance [14]. Furthermore, the emergence of penicillin- and cephalosporin-resistant strains has made treatment more difficult, and a consequence of this is the urgent need for PCVs that are effective in infants. Disparities between races, ages, genders, those living in urban or rural districts, and even socioeconomic situations significantly influence variations in carriage, invasive pneumococcal diseases (IPD), resistance, and even coverage of the available PCVs [15,16,17]. Other studies on the impact of vaccination on racial disparities of IPD incidence showed a high reduction after the introduction of PCV7 and PCV13 [18]. However, pneumococcal studies published for Jordan concentrated only on rural areas, especially where no PCVs were available [1,5,19]. Furthermore, Jordan is a mixture of cultures, especially after the Gulf and Syrian wars, in addition to being home to a large number of other nationalities and refugees. This study aims to differentiate between the urban city of Amman and the rural area of eastern Madaba to evaluate the prevalence of pneumococcal carriage, its resistance, and the coverage of available PCVs. Furthermore, this study seeks to estimate the potential of PCV use in reducing vaccine-type isolates and to indirectly determine the impact of PCVs on disease. We also consider that all PCVs have been found privately on the market since 2000; however, none of these is included in Jordan’s National Immunization Program (NIP).

## 2. Materials and Methods

Ethical clearance statement: The research project was approved by the Independent Ethical Committee (IEC) from the Ministry of Health (MOH) in Jordan, followed by approval by the MOH, as well as by approvals of the directorates of each day care center (DCC) taking part in this research project. Informed written consent was obtained from parents in the form of written permission allowing their children to participate and use NP-swabs; any other relevant data were also used for protocoling. The parents were educated on the benefits of the future vaccination with the available pneumococcal conjugate vaccines. Questionnaires with names, dates of birth, genders, numbers in household, addresses, and history of PCV vaccination were registered at the time of sample collection. All NP samples were collected by trained medical doctors from each DCC. Positive results of carriage with resistance analysis and serotyping were sent to the medical doctors from each DCC in order to be registered on the files of the participating children.

Study population and Design: The estimated population count of Madaba during the study was 199,500, while the estimated population in Amman was 4,226,100; these figures were taken from Jordan’s department of statistics in 2017. Eastern Madaba and the center of Amman were chosen for this study. Nasopharyngeal swabs were taken from outpatients of private clinics in Amman and from governmental day care centers in east Madaba during routine check-ups. Between April 2015 and April 2019, 682 NP-swabs were tested, including 267 samples from the urban area of Amman and 415 samples from the rural area of eastern Madaba.

Laboratory procedures: Single NP swabs were obtained from children. Children involved in the study had not consumed antibiotics in the three months prior to the NP swabs being taken. Processing of NP swabs took place at the microbiological labs of the German Jordanian University. Cultivation on Columbia Agar plates supplemented with 5% sheep blood for identification was enacted, as previously described [5,20]. The plates were incubated at 35 °C with 5% CO_2_ overnight. Identification was performed by conventional microbiological methods, such as colony morphology, susceptibility to optochin (bioMérieux), and bile solubility. Confirmed *S. pneumoniae* isolates were tested for minimal inhibitory concentrations (MIC) using the micro broth dilution method, as recommended by the Clinical Laboratory Standards Institute (CLSI) [21], and using the VITEK2 compact system with Vitek cards (AST03) and an E-test provided by bioMérieux. Antibiotics used were penicillin G (PEN), amoxicillin (AMOX), cefotaxime (CETA), cefuroxime (CEFU), cefpodoxim (CEPO), clarithromycin (CLA), clindamycin (CLI), tetracycline (TET), levofloxacin (LEV), moxifloxacin (MOX), telithromycin (TEL), trimethoprim/sulfamethoxazole (SXT), chloramphenicol (CHA), and vancomycin (VAN). All of these antibiotics were available in the AST03 cards of the VITEK2 compact system. An E-test provided by bioMérieux was used to assure the resistance of isolates. Breakpoints and interpretation of susceptibility were calculated according to the latest CLSI standards. *S. pneumoniae* ATCC 49619 was used as a control strain. Serotyping of the pneumococcal isolates was performed by the Neufeld’s Quellung reaction method using type and factor sera provided by the Statens Serum Institute (SSI), Copenhagen, Denmark.

Statistical analysis: A Student’s *t*-test was considered for significant differences using 2-tailed values with the level of significance at *p* < 0.05. Other analysis included rate of carriage, vaccine and non-vaccine serotype coverage, and resistance rates to antibiotics.

## 3. Results

A total of 682 children aged 1 to 163 months old were enrolled this study. Table 1 describes the gender distribution of all enrolled children, with a total carriage rate in both cities of 36.2%. The total carriage rate for Amman was 31.1%; male carriage rate was 30.3% and female carriage rate was 32.1%. As a comparison, the carriage rate for eastern Madaba was 39.5%; male carriage rate 42.9% and female carriage rate was 35.6%. The difference in female carriage rates between both areas showed no significance, with *p* value equal to 0.626. Significant difference (*p* < 0.05) was noticed for carriage rates in both locations.

Table 2 shows the number of PCV7 injections taken by children from both areas, taking into consideration that all nasopharyngeal samples were taken at least 3 months post-PCV injections in all cases from both urban and rural areas. The number of cases taken from Amman with no history of PCV injections was 198, and 69 (25.8%) cases had a history of 1 to 3 previous injections with PCV7. Only 11 from the 69 (15.9%) vaccinated cases from Amman showed carriage with non-PCV7 serotypes. The number of cases taken from Amman with no history of PCV7 injections was 198, where 72 cases were carriers. As such, 35/72 (48.6%) could be covered by the PCV7, 39/72 (54.2%) could be covered by the PCV13, and 47/72 (65.3%) could be covered by the future PCV20 vaccine. The number of cases that still carried pneumococcus after the first, second, and third injections with PCV7 was 1/69, 3/69, and 7/69, respectively. All serotypes recovered from these cases after previous injections were 6A, 9N, 11A, 23A, and others not included in PCV serotypes. In the rural area of east Madaba, only eight cases had a previous history of two PCV7 injections, where three out of the eight (37.5%) were carriers after the second injections with the serotypes 6B, 19A, and others not included in the PCV7 serotypes. However, there were 407 cases with no history of PCV injection in eastern Madaba; 161/407 (39.6%) cases were carriers. As such, 81/161 (50.3%) could be covered by the PCV7, 97/161 (60.2%) could be covered by the PCV13, and 111/161 (68.9%) could be covered by the future PCV20 vaccine.

Table 3 shows the antibiotic resistance profile for 14 antibiotics for isolates from both areas. MIC criteria for penicillin resistance are the same as those for non-meningitis, i.e., ≥0.12 µg/mL. No resistance was detected for moxifloxacin, levofloxacin, telithromycin, or vancomycin. Resistance to penicillin, clarithromycin, and trimethoprim-sulfamethoxazole was high in both areas. Nevertheless, significant differences (*p* < 0.05) in resistance to penicillin, clarithromycin, clindamycin, and trimethoprim-sulfamethoxazole were higher in the rural area of eastern Madaba than isolates were in the urban area of Amman.

Figure 1 shows the percentage of serotypes detected in each area, along with the rate of occurrence of each serotype. Other serotypes in the figure are those isolated only once and not included in the PCVs. Only one isolate from each carrier was serotyped, and no multiple carriage was detected in this study, as one single NP-swab was taken from each case. Four serotypes of the PCV7 isolated from eastern Madaba (14, 18C, 19F, 23F) showed significantly higher (*p* < 0.05) rates of isolation than in the urban area of Amman, with the exception of the serotype 6B, which was higher in Amman, even with fewer isolates. One strain was isolated from each area with serotype 9V, and serotype 4 was not detected in both areas that were under the remit of PCV7.

Coverage rates of the pneumococcal isolates in both areas are presented in Table 4. Coverage rates for PCV7, PCV10, PCV13, and the future PCV20 were 42.2%, 42.2%, 48.2%, and 60.2% for Amman, respectively. Significantly higher coverage was detected for PCV13 (*p* value = 0.041) and PCV20 (*p* value = 0.043) vaccines in the eastern area of Madaba, with rates of coverage of PCV7 (50%), PCV10 (50%), PCV13 (62.2%), and the future PCV20 (73.2%).

Serotype-related antibiotic resistance in the urban area of Amman in Table 5 clearly shows the highest resistance rates among the serotypes available in the pneumococcal conjugate vaccines, so that the resistance rates for penicillin in serotypes 19F, 6B, 23F, 14, and 9V are 92.3%, 90.9%, 100%, 100%, and 100%, respectively. Similarly, in Table 6, for the rural area of eastern Madaba, 100% resistance rates were detected for penicillin in serotypes 19F, 6B, 23F, 14, 9V, and 18C.

## 4. Discussion

Disparities in pneumococcal infections and carriage were studied worldwide in populations of different socioeconomic statuses, ethnicities, and races [22]. Other studies have been performed on carriage rates before or after the introduction of PCVs [23]. Differences in pneumococcal carriage rates, pneumococcal conjugate vaccine coverage, and resistance rates worldwide depend on the use of vaccination, correct use of antibiotics, socio-economic factors, age, and urban or rural residencies of people [24]. Furthermore, gender and racial disparities in carriage rates were studied in different parts of the world, showing either significant changes or no significant differences [16,25,26]. On the other hand, research conducted on both urban and rural differences in the pneumococcal disease or colonization, with significant differences was rarely found [27,28,29]. This study showed a total carriage rate for both areas of 36.2%, but it showed higher and significant rate of carriage in eastern Madaba (*p* < 0.05) compared to the urban area of Amman. In a study performed in urban and rural areas of Pakistan, carriage rates were more than 70% in children enrolled, and there were no significant differences between the areas [28]. Carriage rates in both eastern Madaba and Amman were much higher than carriage rates obtained from Saudi Arabia (13%) or from Taiwan (15.3%) [30]. Another study conducted on pneumococcal carriages among urban and rural Vietnamese school children showed no difference between urban and rural (suburban) areas, with a 21–22% carriage rate; non-susceptibility to penicillin and erythromycin in both areas was equally distributed [29]. These differences in carriage rates worldwide were related to certain socio-economic conditions, including housing, access to health care, poor hygiene, family size, overcrowded living conditions, day-care contact, and number of siblings [31]. Regarding the situation in Jordan, and in accordance with the historical shift and transformation of rural into urban areas, urbanization is envisioned accelerate and rural areas will be minimized. This force of urbanization will have an impact on reductions in the carriage and resistance. However, further actions should be taken to tackle the other factors affecting carriage, namely unequal socioeconomic situations, represented by high density of housing, as well as other factors. In other words, interdisciplinary strategies should be used to create a better environment that prevents such an impact.

Pneumococcal drug resistance in Jordan has reached very high levels, especially in terms of penicillin, trimethoprim-sulfamethoxazole, and erythromycin in eastern Madaba, with rates of 95.8%, 73.2%, and 68.9%, respectively. Although these levels of resistance were significantly higher in eastern Madaba than in the urban area of Amman, resistance rates of these antibiotics detected in the urban area of Amman were also relatively high. Despite the fact that the pneumococcal conjugate vaccines (PCVs) are not yet available in Jordan in the National Immunization Program (NIP), they are available on the private market. In this study, in the urban area of Amman, 69 of 267 (25.8%) children have received between 1 and 3 injections of PCV7 privately, out of which only 11 (15.9%) cases of the vaccinated children were carriers with serotypes not available in the PCV7. As a comparison with non-vaccinated children in the urban area of Amman, carriage rate was 36.4%. For the rural area of Madaba, only 8 cases from 415 (1.9%) have received two injections of PCV7; three of them were carriers and one case was a carrier of 6B included in the PCV7. The main reasons related to these differences are family income and awareness of vaccination benefits. Internationally, PCVs proved to be highly efficient in preventing serious disease caused by serotypes in the vaccine, and it prevents symptomless colonization of the nasopharynx [19,32]. Prevention of NP colonization in the infection cycle reduces the chances of the infection’s spread, and indirectly protects from disease. Through these indirect effects, the protection afforded by a vaccine extends to the whole population, including those who are not vaccinated (herd protection). As mentioned before, the pneumococcal conjugate vaccines are not yet part of the National Immunization Program (NIP) of Jordan, although an urgent need for PCVs was discussed and approved at the MOH of Jordan. The SARS-CoV-2 crisis worldwide has delayed taking decisions to include the PCV in the NIP of Jordan, and most probably has even delayed the introduction of the new PCV20 from Pfizer because of the mass need for COVID-19 vaccines. The resistance situation of colonizing pneumococcal strains isolated from children was previously studied in other rural areas of Jordan. The rural area of Ajlun, for example, was studied in 2009–2010 and showed a penicillin resistance rate of 84.0% [19]. This rate was found to be 80.0% in 2014 for Wadi Alseer [1], a rural area on the peripheries of Amman, as well as 86.3% in the rural area of eastern Irbid in 2019 and 94.4% in all regions of Madaba between 2017 and 2019 [5]. Resistance rates found in this study for penicillin were 95.8% in the rural area of eastern Madaba, significantly higher (*p* < 0.05) than all rural areas tested before in Jordan. The same was true for clarithromycin, clindamycin, and trimethoprim-sulfamethoxazole, all of which were significantly higher (*p* < 0.05) in eastern Madaba than in the urban area of Amman, as well as being higher than in other rural areas studied in Jordan [1,19]. The main reason for the increase in antibiotic resistance is presumably high intake and consumption of antibiotics without a prescription from a medical doctor [33]. High consumption of antibiotics in the country, and a history of antibiotic consumption prior to their visits to the DCC, could be the reason or could contribute to increasing numbers of resistant strains [33]. In Jordan, there is a shortage of data and publications related to *Streptococcus pneumoniae*, especially in relation to invasive pneumococcal diseases; therefore, no available serotypes of invasive or non-invasive diseases caused by pneumococcus are widely usable. For this reason, monitoring the changes in serotypes of pneumococcal carriage is a practical way to assess vaccine impact. In this study, serotypes available in the PCVs were more significant (*p* < 0.05) in the rural area of eastern Madaba than in the urban area of Amman, as the case for serotypes 3, 6A, 9V, 14, 18C, 19A, 19F, and 23F, with the exception of 6B, shows. This is due to the much higher vaccination rates in Amman than in eastern Madaba, and the exceptional increase in 6B in Amman is mainly caused by non-vaccinated cases. These data are consequent with the low coverage data for all PCVs in Amman compared to eastern Madaba, since vaccination leads to decrease coverage of PCVs. The proportion of circulating pneumococci with serotypes covered by the vaccine may vary among regions. Methodological differences in carriage studies and variations of carriage estimates may result from differences in climate, seasons, or crowding, all of which influence transmission [34]. The rate of antibiotic resistance for vaccine and non-vaccine serotypes, for both areas, showed higher resistance rates for the rural area than Amman. For instance, six serotypes showed no resistance to penicillin for Amman, whereas only one of the serotypes isolated from the rural area of eastern Madaba showed no resistance. These results were consistent with the findings published from Vietnamese urban and rural areas among schoolchildren [29].

This study has some limitations, such as accessing the files of outpatients to collect data, which might help in ascertaining the reasons behind the high rates of carriage and resistance in a scientific and reasonable way. Another difficulty was the transportation of samples collected or the delay in delivery of the samples to the lab, both of which affect the viability of the isolates and achieve higher rates of carriage.

## 5. Conclusions

The disparity of pneumococcal carriage, resistance, and even the coverage of pneumococcal conjugate vaccines between urban and rural areas for Jordanian children was significant. In contrast to theoretical discourses and debates all over the world that show little disparity between rural and urban isolates, there is a significant and profound value to be found in comparing rural and urban communities in Jordan. As rural local communities have different lifestyles to their urban counterparts, patterns of carriages relate to the origin of their carriers. Socio-economic status, along with social habits and norms, has proven to have a valid effect on increasing carriages in rural areas, with great disparities in urban areas. As a present research topic in Jordan, this research has pinpointed the importance of continuous prevalence studies within local communities to showcase locally oriented differentiations in regard to rural/urban impacts. The introduction of the pneumococcal conjugate vaccine into the National Immunization Program of the country would make a substantial change to such differences. These data prove the significant disparities between urban and rural areas of Jordan when it comes to carriage, resistance, serotype distribution, and coverage of PCVs; hopefully, these findings are indicative of better solutions in the future.

## Figures and Tables

**Figure 1 vaccines-09-00789-f001:**
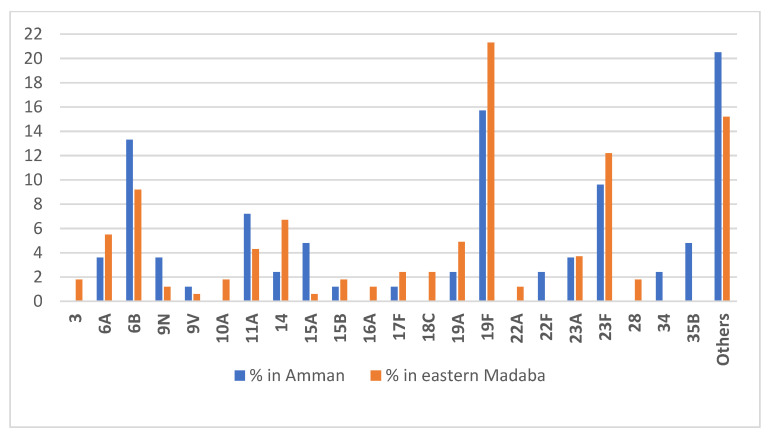
Serotype distribution of isolates from Amman and eastern Madaba.

**Table 1 vaccines-09-00789-t001:** Source of NP samples according to the gender and total carriage, 2015–2019.

City	No. Samples	Male *n* (%)	Male Carrier *n* (%)	Female *n* (%)	Female Carrier *n* (%)	Total Carriage *n* (%)
Amman	267	155 (58.1)	47 (30.3)	112 (41.9)	36 (32.1)	83 (31.1)
East Madaba	415	224 (54.0)	96 (42.9)	191 (46.0)	68 (35.6)	164 (39.5)
Total	682	379 (55.6)	143 (37.7)	303 (44.4)	104 (34.3)	247 (36.2)

Note: Statistical significance conducted depending on *p* value. Significant at *p* < 0.05 level.

**Table 2 vaccines-09-00789-t002:** Carriers from Amman and East Madaba with a history of PCV7 injections.

City	# PCV7 Injections	Carriage *n* (%)	* Serotype or VTs or NVTs Recovered
Amman *n* = 267	1 injection (*n* = 11)	1/11 (9.1%)	6A
	2 injections (*n* = 9)	3/9 (33.3%)	All NVT
	3 injections (*n* = 49)	7/49 (14.3%)	All NVT
	0 injections (*n* = 198)	72/198 (36.4%)	VTs (*n* = 35) (48.6%)
Eastern Madaba *n* = 415	1 injection (*n* = 0)	0/0 (0.0%)	none
	2 injections (*n* = 8)	3/8 (37.5%)	6B; 19A; others
	3 injections (*n* = 0)	0/0 (0.0%)	none
	0 injection (*n* = 407)	161/407 (39.6%)	VTs (*n* = 84) (52.2%)

* VTs and NVTs = Vaccine types and non-vaccine types related to PCV7.

**Table 3 vaccines-09-00789-t003:** Comparison of antibiotic resistance in both Amman and eastern Madaba.

Antibiotic	Amman (*n* = 83/267)				Eastern Madaba (*n* = 164/415)			
	%S	%I + R	MIC_50_	MIC_90_	% S	% I + R	MIC_50_	MIC_90_
Penicillin	18.1	81.9	1	4	4.3	95.8	1	2
Amoxicillin	85.5	14.5	1	>4	92.7	7.3	0.5	2
Cefotaxime	96.4	3.6	0.5	1	93.9	6.1	0.5	1
Cefuroxime	41.0	59.0	4	>4	39.0	61.0	2	>4
Cefpodoxime	39.8	60.2	1	2	35.4	64.6	1	2
Clarithromycin	41.0	59.0	2	>32	31.1	68.9	8	>32
Clindamycin	68.7	31.3	≤0.125	>32	59.2	40.8	0.06	>32
Moxifloxacin	100	0.0	0.125	0.25	100	0.0	0.125	0.25
Levofloxacin	100	0.0	1	2	100	0.0	1	2
Trimethoprim- Sulfamethoxazole	38.6	61.4	2/38	>8/152	26.8	73.2	4/76	>8/152
Tetracycline	48.2	51.8	8	16	48.8	51.2	4	32
Chloramphenicol	92.8	7.2	≤4	≤4	96.3	3.7	≤4	≤4
Telithromycin	100	0.0	0.03	0.125	100	0.0	0.016	0.06
Vancomycin	100	0.0	0.5	0.5	100	0.0	0.5	0.5

**Table 4 vaccines-09-00789-t004:** Coverage of *Streptococcus pneumoniae* to pneumococcal conjugate vaccines.

Coverage in Amman and Eastern Madaba	PCV7 *n* (%)	PCV10 *n* (%)	PCV13 *n* (%)	PCV20 *n* (%)
Amman (*n* = 83)	35 (42.2)	35 (42.2)	40 (48.2)	50 (60.2)
Eastern Madaba (*n* = 164)	82 (50.0)	82 (50.0)	102 (62.2)	120 (73.2)

**Table 5 vaccines-09-00789-t005:** Serotype-related antibiotic resistance to isolates from Amman.

Serotypes from Amman	% Pen R	% Cla R	% Cli R	% Lev R	% Sxt R	% Tet R	% Cha R
6A (*n* = 3)	100%	66.7%	66.7%	0.0%	33.3%	66.7%	0.0%
6B (*n* = 11)	90.9%	72.7%	72.7%	0.0%	90.9%	63.6%	18.2%
6C (*n* = 1)	100%	100%	100%	0.0%	0.0%	100%	0.0%
7B (*n* = 1)	100%	100%	0.0%	0.0%	100%	100%	0.0%
9N (*n* = 3)	100%	0.0%	0.0%	0.0%	33.3%	100%	0.0%
9V (*n* = 1)	100%	100%	0.0%	0.0%	100%	100%	0.0%
11A (*n* = 6)	100%	83.3%	16.7%	0.0%	100%	66.7%	0.0%
14 (*n* = 2)	100%	100%	100%	0.0%	100%	100%	0.0%
15A (*n* = 4)	25%	100%	75.0%	0.0%	100%	100%	0.0%
15B (*n* = 1)	100%	0.0%	0.0%	0.0%	0.0%	0.0%	0.0%
15F (*n* = 1)	100%	0.0%	0.0%	0.0%	0.0%	0.0%	0.0%
16F (*n* = 1)	100%	100%	0.0%	0.0%	0.0%	0.0%	0.0%
17F (*n* = 1)	0.0%	0.0%	0.0%	0.0%	0.0%	0.0%	0.0%
18A (*n* = 1)	0.0%	0.0%	0.0%	0.0%	0.0%	0.0%	0.0%
19A (*n* = 2)	100%	100%	50%	0.0%	50%	50%	0.0%
19F (*n* = 13)	92.3%	84.6%	46.2%	0.0%	92.3%	76.9%	0.0%
22F (*n* = 2)	0.0%	0.0%	0.0%	0.0%	0.0%	0.0%	0.0%
23A (*n* = 3)	66.7%	33.3%	33.3%	0.0%	33.3%	33.3%	0.0%
23F (*n* = 8)	100%	50%	0.0%	0.0%	100%	50%	0.0%
33A (*n* = 1)	0.0%	0.0%	0.0%	0.0%	0.0%	0.0%	50.0%
33F (*n* = 1)	0.0%	0.0%	0.0%	0.0%	0.0%	0.0%	0.0%
34 (*n* = 2)	50%	0.0%	0.0%	0.0%	0.0%	0.0%	0.0%
35B (*n* = 4)	100%	0.0%	0.0%	0.0%	0.0%	0.0%	0.0%
35A (*n* = 1)	0.0%	0.0%	0.0%	0.0%	100%	0.0%	0.0%
NT (*n* = 1)	100%	100%	0.0%	0.0%	100%	100%	0.0%
Others (*n* = 8)	87.5%	62.5%	12.5%	0.0%	12.5%	12.5%	0.0%

**Table 6 vaccines-09-00789-t006:** Vaccine and non-vaccine serotype-related antibiotic resistance to isolates from eastern Madaba.

Serotypes from Eastern Madaba	% Pen R	% Cla R	% Cli R	% Lev R	% Sxt R	% Tet R	% Cha R
3 (*n* = 3)	33.3%	0.0%	0.0%	0.0%	33.3%	33.3%	0.0%
6A (*n* = 9)	100%	100%	66.7%	0.0%	44.4%	66.7%	11.1%
6B (*n* = 15)	100%	73.3%	60.0%	0.0%	86.7%	73.3%	13.3%
6C (*n* = 1)	100%	100%	100%	0.0%	0.0%	100%	0.0%
7B (*n* = 1)	100%	100%	100%	0.0%	0.0%	100%	0.0%
9N (*n* = 2)	100%	0.0%	0.0%	0.0%	50.0%	100%	0.0%
9V (*n* = 1)	100%	0.0%	0.0%	0.0%	100%	0.0%	0.0%
10A (*n* = 3)	66.7%	0.0%	0.0%	0.0%	66.7%	0.0%	0.0%
11A (*n* = 7)	100%	42.9%	14.3%	0.0%	71.4%	28.6%	0.0%
14 (*n* = 12)	100%	100%	100%	0.0%	90.9%	63.6%	0.0%
15A (*n* = 1)	100%	100%	0.0%	0.0%	100%	100%	0.0%
15B (*n* = 3)	100%	100%	0.0%	0.0%	66.7%	100%	0.0%
15C (*n* = 1)	100%	100%	0.0%	0.0%	100%	100%	0.0%
16A (*n* = 2)	50%	0.0%	0.0%	0.0%	50%	0.0%	0.0%
16F (*n* = 1)	100%	0.0%	0.0%	0.0%	100%	0.0%	0.0%
17F (*n* = 4)	100%	0.0%	0.0%	0.0%	25.0%	0.0%	0.0%
18C (*n* = 4)	100%	0.0%	0.0%	0.0%	75%	25.0%	0.0%
19A (*n* = 8)	100%	100%	25.0%	0.0%	87.5%	25.0%	0.0%
19F (*n* = 35)	100%	97.1%	82.9%	0.0%	94.3%	91.4%	0.0%
22A (*n* = 2)	100%	0.0%	0.0%	0.0%	0.0%	0.0%	0.0%
23A (*n* = 6)	100%	66.7%	16.7%	0.0%	50.0%	33.3%	0.0%
23B (*n* = 1)	100%	0.0%	0.0%	0.0%	0.0%	0.0%	0.0%
23F (*n* = 20)	100%	43.8%	6.3%	0.0%	81.3%	37.5%	12.5%
28A (*n* = 3)	100%	100%	66.7%	0.0%	100%	0.0%	0.0%
35A (*n* = 1)	100%	100%	0.0%	0.0%	100%	0.0%	0.0%
Pool C (*n* = 1)	0.0%	0.0%	0.0%	0.0%	0.0%	0.0%	0.0%
Pool G (*n* = 5)	100%	100%	80.0%	0.0%	80.0%	20.0%	20.0%
Pool I (*n* = 1)	100%	100%	0.0%	0.0%	100%	100%	0.0%
NT (*n* = 5)	60.0%	80.0%	0.0%	0.0%	100%	20.0%	0.0%
Others (*n* = 7)	100%	28.6%	28.6%	0.0%	14.3%	28.6%	0.0%

## Data Availability

Not applicable.

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
