# Peer review of "Urban and Rural Disparities in Pneumococcal Carriage and Resistance in Jordanian Children, 2015–2019"

_vaccines, 2021, doi:10.3390/vaccines9070789_

Round 1

Reviewer 1 Report

In the manuscript "Urban and rural disparities in pneumococcal carriage and resistance of Jordanian children, 2015-2019” Adnan Al-Lahham and colleagues determined differences in pneumococcal carriage rate, resistance, and serotypes from healthy Jordanian children of Amman (urban) and eastern Madaba (rural).

I guess that the aim of the study is basically to evaluate the pneumococcal carriage prevalence to estimate the potential for the use of PCVs in reducing the vaccine type isolates and indirectly determine the impact of the PCVs on disease in order to strongly recommend an immunization program although it’s only privately available.

They found significant differences in carriage, resistance and coverage in both regions and that the potential inclusion of PCV vaccination program for rural areas is essential.

In order to make clearer the message of the MS I would suggest to put more attention in reporting data and results.

In particular, authors propose only tables, I would suggest to switch some of them into graph or panel (for example the may show the serotypes’ resistance data as a color-coded panel, as well as the serotypes distribution).

I would suggest to show the most interesting data in a better way in order to strengthen the bottom line of the message and the whole MS.

Here just some comments (among many others) :

Lin 37: coma instead full stop? The sentence is truncated.

Line 43: “cigarette smoking” In healthy children?

Line 51: what the authors mean with “introduction of PCVs proved to have excellent efficacy NP- 51 carriage and resistance”?

Line 175-176: the authors stated that “This study showed a total carriage rate for both areas of 36.2%, but it showed higher and significant rate of carriage in eastern Madaba (P<0.05) compared to the urban area of Amma” in contrast to other studies that did not find any differences between rural and urban carriage. The authors hastily assert it is due “to certain socio-economic conditions including 185 housing, access to health care, poor hygiene, family size, overcrowded living conditions, day-care contact, and number of siblings (31)” without adding any new information or speculation then that already reported in general.

Moreover, I would really appreciate they discuss some of the limitations of the study, which would highlight a strong analytical sense and scientific responsibility, meaning that they got the weaknesses of the study and its design, I would suggest to rewrite the MS in order to get clear and more convincing narrative.

Author Response

Response to Reviewer 1 comments:

Thank you so much for the manuscript entitled" Urban and rural disparities in pneumococcal carriage and resistance of Jordanian children, 2015-2019."

Here are your review with the responses: 

Point 1: In the manuscript "Urban and rural disparities in pneumococcal carriage and resistance of Jordanian children, 2015-2019” Adnan Al-Lahham and colleagues determined differences in pneumococcal carriage rate, resistance, and serotypes from healthy Jordanian children of Amman (urban) and eastern Madaba (rural). I guess that the aim of the study is basically to evaluate the pneumococcal carriage prevalence to estimate the potential for the use of PCVs in reducing the vaccine type isolates and indirectly determine the impact of the PCVs on disease in order to strongly recommend an immunization program although it’s only privately available.

Aims have been adjusted accordingly and were added in the aims paragraph 

They found significant differences in carriage, resistance and coverage in both regions and that the potential inclusion of PCV vaccination program for rural areas is essential.

Point 2: In order to make clearer the message of the MS I would suggest to put more attention in reporting data and results.

The manuscript was given more attention and more corrections were done in reporting the data and results

Point 3: In particular, authors propose only tables, I would suggest to switch some of them into graph or panel (for example the may show the serotypes’ resistance data as a color-coded panel, as well as the serotypes distribution).

I would suggest to show the most interesting data in a better way in order to strengthen the bottom line of the message and the whole MS.

In this regard, table 4 was changed to a figure, so that the reader would have a fast look at the differences, but some of the serotypes were taken out (since they are available in the last two tables) to make the figure fitting in the text. Serotypes taken out from the table were only isolated once and were not part of the PCVs 

Point 4: Line 37: coma instead full stop? The sentence is truncated.

The sentence was corrected accordingly

Point 5: Line 43: “cigarette smoking” In healthy children?

Family smoking in members of the family was added

Point 6: Line 51: what the authors mean with “introduction of PCVs proved to have excellent efficacy NP- 51 carriage and resistance”?

The statement was changed with a reference to: 

Therefore, introduction of PCVs was necessary as they proved to be efficient in decreasing NP-carriage and resistance (14). 

Point 7: Line 175-176: the authors stated that “This study showed a total carriage rate for both areas of 36.2%, but it showed higher and significant rate of carriage in eastern Madaba (P<0.05) compared to the urban area of Amman” in contrast to other studies that did not find any differences between rural and urban carriage. The authors hastily assert it is due “to certain socio-economic conditions including 185 housing, access to health care, poor hygiene, family size, overcrowded living conditions, day-care contact, and number of siblings (31)” without adding any new information or speculation then that already reported in general.

New information and speculation were added to the manuscript as requested 

This paragraph was added as a response: 

About the situation in Jordan, and in accordance to the historical shift and transformation of rural into urban areas, the urbanization process is envisioned to be accelerated and the rural areas will be minimized. As a main factor of this carriage and resistance, this force of urbanization will have an impact on the reduction of the carriage and resistance. However, further actions should be made to tackle the other factors affecting the carriage namely, the socioeconomic situation that is represented in the high density of housing among other factors. In other words, interdisciplinary strategies shall take place in creating a better environment that prevents such impact.

Point 8: Moreover, I would really appreciate they discuss some of the limitations of the study, which would highlight a strong analytical sense and scientific responsibility, meaning that they got the weaknesses of the study and its design, I would suggest to rewrite the MS in order to get clear and more convincing narrative.

A last paragraph was added for the study limitations and was inserted before the conclusions, which have to do with the access to outpatients data and some times the delay in transportations of samples

Thank you so much 

Reviewer 2 Report

Article describing the pneumococcal carriage in Jordan children during 4 years comparing serotypes and resistance patterns between carriage in rural and urban areas with different percentages of PCV7 populations.

Minor comments.

Introduction

Line 35. I suggest changing “sickness” with “diseases” or “infections”.

Lines 35-37. There is no verb in the first sentence.

Line 39. “…nasopharynx during the first few months of life…” I suggest adding “…nasopharynx mainly during the first few months of life…”

Line 48. Write S. pneumoniae in italics and all though the text.

Line 49-50. Please, include the year of the WHO estimation.

Line 49-52. Please, rewrite the sentence in a more comprehensive way or divide the sentence in two.

Methods.

  • Where (in which Laboratory) were microbiological procedures performed? Please include in the text.
  • Study population. Please indicate, if known, the children population for each region in the years of the study to have an idea of the representativeness of the sample.
  • Pneumococcal isolation. The authors refer the cultures of NP swabs to the work of Watt JP. et al (reference 20). However, the procedure of ref. 20 is for adult population. They also do oropharyngeal collection and more important, use a gentamicin blood agar plate for the selection of pneumococcal isolates. They also used the recommended STGG medium for swab transport. Where all these procedures used in the study? If so, please include briefly in methods.
  • How many isolates were serotyped from each child? Pneumococcal multiple carriage is common in young children. Please point out in the text the number of isolates serotyped from each sample.
  • Had the children received antibiotics in the moths previous to the sample collection? Please point out in the text.
  • Antimicrobial susceptibility. What antibiotics were studied by VITEK and which by E-test?
  • What was the MIC criteria used for penicillin resistance (meningitis ≥ 0.12 µg/mL non-meningitis oral penicillin ≥ 2 µg/mL)? Please sate it in the results section.
  • Statistical analysis. I am not sure that the t-test is correct for analyzing categorical variables as is the case in this study.

Results

Lines 106-108. “The study 106 includes the urban area of Amman as the capital city (n=267), and the rural area of eastern 107 Madaba with 415 children”. This is already stated in lines 83-84.

Line 112. The difference in female carriage from different areas is not significant (The two-tailed P value equals 0.626). This should be stated in results.

Table 2. NVT could be deleted as can be deduced from VT found.

Line 150-154 and table 5. The significant differences (p<0.05) in VT in both areas were for all PCV vaccines? Applying the Fisher exact text, p values for PCV7 and PCV10 coverage was 0.281, for PCV13 was 0.041 and for PCV20 was 0.043

Tables 6 and 7. Resistance of serotypes with only 1 or 2 isolates is not very representative. Due to the high number of tables, may be the authors can group serotype resistance percentages in PCV7, PCV13 (or the extra 6 serotypes of PCV13 not included in PCV7: serotypes 1, 3, 5, 6A, 7F, 19A), and PCV20 (or extra non-PCV13 serotypes)  comparing both regions. 

Discussion.

Lines 179-180. Please, include the % or of carriage in Saudi Arabia and Taiwan

Lines 189 and 218. It is different penicillin resistance that non-susceptible, as intermediate-susceptible isolates can be considered as non-resistant depending on the penicillin dose used.

Line 207. I recommend using “herd protection” instead of “herd immunity” as immunity is not provided at such.

Line 210. I suggest changing “corona crisis” with “SARS-CoV-2 crisis”

Line 218. I suggest specifying: “…was found to be 95.8% in the rural are of Madaba as significantly higher…” because in Amman resistance was 81.9%. Besides, in reference 5, from the same author as this study, penicillin resistance rate in Madaba was 94.4% for 2017-2019. This could be included in the discussion.

Line 221. “… was the highest in eastern Madaba compared to other rural areas studied in Jordan”. Please, include reference for this statement

Lines 223-224. “These resistance rates were variable among DCCs and in each season.”.

There are no results showing differences between DCC or seasons.

Lines 223-225. “These resistance rates were variable among DCCs and in each season. High consumption of antibiotics in the country, and a history of antibiotic consumption prior to their visits to the DCC could be the reason or contribute to increased resistant strains (34, 35).” These two lines are identical (including references) to those written by the author in the discussion of reference 5.

Line 233 “This is due to the much higher vaccinated cases from Amman than in eastern Madaba” and Line 235: “These data are consequent with the low coverage data for all PCVs in Amman compared to eastern Madaba.” Aren’t these two sentences somehow contradictory?

Line 252. Lift styles or Lifestyles?

Lines 255-257. “As a precedent research topic in Jordan, with rear reference of similar surveys, this research has pinpointed the aspect of envisioning typologies of isolates within local communities to show case local oriented differentiations in regard to rural/urban impacts.”

I am not sure to fully understand this sentence. I suggest rewriting it in a clearer way.

Author Response

Responses to Reviewer 2 Comments: 

Minor comments.

Introduction

Point 1: Line 35. I suggest changing “sickness” with “diseases” or “infections”.

The word was changed accordingly

Point 2: Lines 35-37. There is no verb in the first sentence.

Verb was added as requested

Point 3: Line 39. “…nasopharynx during the first few months of life…” I suggest adding “…nasopharynx mainly during the first few months of life…”

The sentence was changed as requested

Point 4: Line 48. Write S. pneumoniae in italics and all though the text.

All S. pneumoniae were changed to italics as requested

Point 5: Line 49-50. Please, include the year of the WHO estimation.

The year was added 2006

Point 6: Line 49-52. Please, rewrite the sentence in a more comprehensive way or divide the sentence in two.

done as requested

Methods.

Point 7: Where (in which Laboratory) were microbiological procedures performed? Please include in the text.

The microbiological lab of the German Jordanian university was added

Point 8: Study population. Please indicate, if known, the children population for each region in the years of the study to have an idea of the representativeness of the sample.

Population counts were added as requested

Point 9: Pneumococcal isolation. The authors refer the cultures of NP swabs to the work of Watt JP. et al (reference 20). However, the procedure of ref. 20 is for adult population. They also do oropharyngeal collection and more important, use a gentamicin blood agar plate for the selection of pneumococcal isolates. They also used the recommended STGG medium for swab transport. Where all these procedures used in the study? If so, please include briefly in methods.

There was a reference mistake, the correct references were added, since we were doing these methods at the National Reference of Streptococci in Germany. 

Point 10: How many isolates were serotyped from each child? Pneumococcal multiple carriage is common in young children. Please point out in the text the number of isolates serotyped from each sample.

There was no multiple carriage in this research project, since only one NP-swab was taken from each child. Multiple carriage was very obvious for other locations as for Ajlun in Jordan, since 3 swabs were taken from each child over one year period. This was added to the text

Point 11: Had the children received antibiotics in the months previous to the sample collection? Please point out in the text.

This point was mentioned in the text as requested, so that no children have received antibiotics in the last 3 months before the sample collection. In some cases it was very difficult to access the files of the children to get such information. This was mentioned as limitations of the study project at the end of the manuscript

Point 12: Antimicrobial susceptibility. What antibiotics were studied by VITEK and which by E-test?

This was mentioned and added in the text. Briefly, all antibiotics were available in the AST03 cards of VITEK2 compact. All resistant isolates were double checked with E tests

Point 13: What was the MIC criteria used for penicillin resistance (meningitis ≥ 0.12 µg/mL non-meningitis oral penicillin ≥ 2 µg/mL)? Please sate it in the results section.

This point was also added in the text for penicillin breakpoints as mentioned in the CLSI standards (Intermediate resistance 0.12 µg/mL-1 µg/mL, high grade resistance 2 µg/mL)

Point 14: Statistical analysis. I am not sure that the t-test is correct for analyzing categorical variables as is the case in this study.

All t-test values mentioned in your review were checked and added in the text (much appreciated)

Results

Point 15: Lines 106-108. “The study 106 includes the urban area of Amman as the capital city (n=267), and the rural area of eastern 107 Madaba with 415 children”. This is already stated in lines 83-84.

This is true and was deleted to avoid repetition 

Point 16: Line 112. The difference in female carriage from different areas is not significant (The two-tailed P value equals 0.626). This should be stated in results.

This was added and stated in the results as requested (much appreciated)

Point 17: Table 2. NVT could be deleted as can be deduced from VT found.

This was deleted as requested (Table accordingly adjusted) 

Point 18: Line 150-154 and table 5. The significant differences (p<0.05) in VT in both areas were for all PCV vaccines? Applying the Fisher exact text, p values for PCV7 and PCV10 coverage was 0.281, for PCV13 was 0.041 and for PCV20 was 0.043

Text was changed only for the PCV13 and PCV20 with the values given (much appreciated)

Point 19: Tables 6 and 7. Resistance of serotypes with only 1 or 2 isolates is not very representative. Due to the high number of tables, may be the authors can group serotype resistance percentages in PCV7, PCV13 (or the extra 6 serotypes of PCV13 not included in PCV7: serotypes 1, 3, 5, 6A, 7F, 19A), and PCV20 (or extra non-PCV13 serotypes)  comparing both regions. 

Table number 4 was changed to a figure with histogram for better showing of the differences (First reviewer response). With regard to tables 6 and 7 (Now 5 and 6) were put to detect the exact serotype resistance for both areas. Your idea is exclusively important will use this idea in my next publication (If you allow). This needs a new analysis for all the PCVs and their found resistance to all antibiotics

Discussion.

Point 20: Lines 179-180. Please, include the % or of carriage in Saudi Arabia and Taiwan

Percentages were added as requested

Point 21: Lines 189 and 218. It is different penicillin resistance that non-susceptible, as intermediate-susceptible isolates can be considered as non-resistant depending on the penicillin dose used.

This is true, since patients with intermediate resistance to penicillin can still be treated with higher doses. In most publications for S. pneumoniae use (I+R) as resistant 

Point 22: Line 207. I recommend using “herd protection” instead of “herd immunity” as immunity is not provided at such.

Done as requested

Point 23: Line 210. I suggest changing “corona crisis” with “SARS-CoV-2 crisis”

Done as requested

Point 24: Line 218. I suggest specifying: “…was found to be 95.8% in the rural are of Madaba as significantly higher…” because in Amman resistance was 81.9%. Besides, in reference 5, from the same author as this study, penicillin resistance rate in Madaba was 94.4% for 2017-2019. This could be included in the discussion.

This was included and added in the discussion as requested

Point 25: Line 221. “… was the highest in eastern Madaba compared to other rural areas studied in Jordan”. Please, include reference for this statement

References were added as requested

Point 26: Lines 223-224. “These resistance rates were variable among DCCs and in each season.”. There are no results showing differences between DCC or seasons.

This statement was deleted (came by mistake)

Point 27: Lines 223-225. “These resistance rates were variable among DCCs and in each season. High consumption of antibiotics in the country, and a history of antibiotic consumption prior to their visits to the DCC could be the reason or contribute to increased resistant strains (34, 35).” These two lines are identical (including references) to those written by the author in the discussion of reference 5.

There was a mistake in the references 34 and 35, since in Jordan the only publication published for "Otoom" mentioned the problem of high antibiotics intake and consumption. Reference was changed

Point 28: Line 233 “This is due to the much higher vaccinated cases from Amman than in eastern Madaba” and Line 235: “These data are consequent with the low coverage data for all PCVs in Amman compared to eastern Madaba.” Aren’t these two sentences somehow contradictory?

This was clarified in the sense that high vaccination in the population leads to decrease vaccine serotypes and consequently leads to decrease coverage of these PCVs. Statement added and corrected 

Point 29: Line 252. Lift styles or Lifestyles?

Changed as requested 

Point 30: Lines 255-257. “As a precedent research topic in Jordan, with rear reference of similar surveys, this research has pinpointed the aspect of envisioning typologies of isolates within local communities to show case local oriented differentiations in regard to rural/urban impacts.” I am not sure to fully understand this sentence. I suggest rewriting it in a clearer way.

The statement was changed as requested

Appreciate all the precious and valuable review points to the manuscript